# COR/LEA Proteins as Indicators of Frost Tolerance in Triticeae: A Comparison of Controlled versus Field Conditions

**DOI:** 10.3390/plants10040789

**Published:** 2021-04-16

**Authors:** Klára Kosová, Miroslav Klíma, Ilja Tom Prášil, Pavel Vítámvás

**Affiliations:** Department of Crop Genetics and Breeding, Crop Research Institute, Drnovská 507/73, 16106 Prague, Czech Republic; Klima@vurv.cz (M.K.); Prasil@vurv.cz (I.T.P.); Vitamvas@vurv.cz (P.V.)

**Keywords:** cold acclimation, vernalisation, frost tolerance, dehydrins, COR14b, growth chambers, field trials

## Abstract

Low temperatures in the autumn induce enhanced expression/relative accumulation of several cold-inducible transcripts/proteins with protective functions from Late-embryogenesis-abundant (LEA) superfamily including dehydrins. Several studies dealing with plants grown under controlled conditions revealed a correlation (significant quantitative relationship) between dehydrin transcript/protein relative accumulation and plant frost tolerance. However, to apply these results in breeding, field experiments are necessary. The aim of the review is to provide a summary of the studies dealing with the relationships between plant acquired frost tolerance and COR/LEA transcripts/proteins relative accumulation in cereals grown in controlled and field conditions. The impacts of cold acclimation and vernalisation processes on the ability of winter-type Triticeae to accumulate COR/LEA proteins are discussed. The factors determining dehydrin relative accumulation under controlled cold acclimation treatments versus field trials during winter seasons are discussed. In conclusion, it can be stated that dehydrins could be used as suitable indicators of winter survival in field-grown winter cereals but only in plant prior to the fulfilment of vernalisation requirement.

## 1. Introduction

Low temperatures represent the dominant environmental factor determining plant growth and development during autumn/winter/spring seasons in temperate climates. Plants have evolved complex mechanisms aimed at an enhancement of their low-temperature tolerance in the process of cold acclimation [1]. In addition, plants growing in temperate climates where regular long periods of cold temperatures occur during the winter season have evolved developmental adaptation preventing a premature transition from more tolerant vegetative stage to cold-susceptible reproductive stage, i.e., complex mechanisms known as “vernalisation” [2]. In Triticeae, two major loci determining the resulting frost tolerance were identified on the long arm of group 5 chromosomes including *Vrn1/Fr1* locus encoding the major vernalisation gene *VRN1* induced by vernalisation fulfilment [3,4] and *Fr2* locus encoding a cluster of cold-inducible *CBF* transcription regulators upstream of *COR/LEA* genes [5,6]. In the field, winter seasons in temperate zones are characterised not only by low temperatures but by multiple environmental stresses including freeze–thaw cycles, temporal floods during snow melting or winter drought. All these environmental stresses induce plant stress responses aimed at an enhancement of stress tolerance. Since low temperatures pose an enhanced risk of cellular dehydration and oxidative stress resulting from imbalances in photosynthesis, induction of enhanced low-temperature tolerance in plants is underlain by enhanced accumulation of several stress-protective proteins including several COR/LEA proteins including several COR/LEA proteins and dehydrins.

Late embryogenesis abundant (LEA) proteins, originally identified in cotton seeds during embryo maturation and desiccation, represent a superfamily of hydrophilic intracellular proteins which accumulate in plant cell under cellular dehydration [7]. Several LEA proteins belonging to either LEA-II (D-11; dehydrins) group characterised by the presence of at least one copy of 15-mer conserved lysine-rich sequence, the K-segment, or LEA-III (D-7/D-29) group characterised by the presence of at least one copy of conserved 11-mer motif, are induced by low temperatures, thus are collectively termed as cold-regulated (COR) proteins [8]. Dehydrins are intracellular hydrophilic proteins which are classified as Group II LEA proteins and are defined by the presence of at least one copy of lysine-rich sequence known as K-segment near the *C*-terminus of their molecule [9,10,11]. In addition, they can also contain a tyrosin-rich sequence (Y-segment) near the *N*-terminus and a stretch of serine residues (S-segment) which could be phosphorylated and function as a nuclear localisation signal [12]. Based on the presence of the conserved motifs, five dehydrin structural groups (Kn, SKn, YxSKn, YxKn and KnS) can be distinguished [11]. Dehydrins are generally characterised by high content of glycine and hydrophilic amino acids and the absence of cysteine and tryptophan. They belong to intrinsically unstructured proteins (IUPs) since they bind relatively large amounts of water molecules and acquire random coils in well-hydrated environments [13,14,15,16]. Under reduced hydration, they form hydrogen bonds between side amino acid residues which result in the formation of defined secondary structures such as α-helices and β-sheets. For example, K-segments form amphipathic α-helices with acidic, basic and hydrophobic sides, thus enabling interactions with both hydrophilic and hydrophobic surfaces such as membranes [17]. Under stress conditions, stress-responsive COR/LEA proteins accumulate to very high levels in plant cells to prevent cellular dehydration; it is estimated that they can reach up to 1–2% of total cellular proteins [18]. Due to the versatility of their structure responding to cell hydration status, dehydrins can adopt multiple molecular functions corresponding to cell hydration status and interaction partners which include not only proteins but also membrane surfaces [15,16]. Studies dealing with dehydrin molecular functions under cold reported their chaperone, cryoprotective, antifreeze, ion-binding and radical scavenging functions for some cold-responsive dehydrins [14,19]; for more details on dehydrin molecular structure and functions, see the work of Kosová et al., [20]. In winter wheat and barley plants, the major cold-inducible dehydrins belong to Kn structural group harbouring multiple K-segments in their molecules (K6 in wheat WCS120 while K9 in barley DHN5) and revealing significant cryoprotective activities [18,19]. At transcript level, cold induction was also reported for acidic SKn type dehydrins including *Dhn8* in barley and three *WCOR410* homologues in wheat [21]. Cold-regulated LEA-III proteins include several chloroplast-targeted proteins regulated by both light and low temperatures such as WCS19, WCOR14a/b, WCOR15, etc. [22,23].

Several studies in controlled conditions (growth chambers) revealed a significant positive relationship (a positive correlation) between COR/LEA (dehydrin) transcript/protein relative accumulation and acquired frost tolerance for some cold-inducible proteins [24,25,26] not only under cold but also under a broad range of growing temperatures from cold to optimum ones [27,28] due to higher threshold induction temperatures of acquired frost tolerance determined as lethal temperature for 50% samples (LT_50_) for highly tolerant genotypes of winter cereals when compared to the less tolerant ones [29]. Some cold-inducible dehydrins, e.g., WCS120 in wheat or DHN5 in barley, can thus be considered as markers of plant acquired frost tolerance for young plants in vegetative stage grown under controlled conditions. However, to be utilised in breeding, field trials are necessary. The results of field experiments indicate that dehydrins could be used as indicators of plant winter survival but only under certain conditions [30].

The aim of the present review is to provide a summary of the recent studies dealing with cold-inducible COR/LEA proteins and their relationships with plant acquired frost tolerance in both controlled and field conditions. Environmental and developmental factors affecting COR/LEA protein relative accumulation are discussed.

## 2. Cold Acclimation Studies in Controlled Conditions

Plants as poikilothermic organisms have to adjust their metabolism to ambient temperatures. Adjustment of chilling-tolerant and freezing-tolerant plants such as Triticeae cereals or winter oilseed rape to low above zero temperatures in the range 0–15 °C resulting in the acquisition of enhanced frost tolerance (FT) is known as cold acclimation (CA) [1,31]. Acquired FT is usually expressed as lethal temperature of 50% of the sample (LT_50_) which can be determined either by a direct frost test including regulated freezing and thawing of the samples in laboratory freezers [32] or by determination of tissue damage via electrolyte leakage [33]. The key inducing factors in plant cold acclimation process represent both low temperature (chilling) and photoperiod (short days) which determine chloroplast redox status and the activity of photosynthetic apparatus which provides energy for biosynthesis of novel molecules of transcripts, proteins and metabolites necessary for plant adaptation to altered environment [34]. CA appears to represent a modular response associated with alterations in nucleosomal histone H2A isoforms, indicating chromatin remodelling resulting in transcriptome and proteome reprogramming [35,36].

Cold acclimation (CA) thus represents a complex process of plant reversible adaptation to low temperatures aimed at enhancement of FT, which is associated with an enhanced accumulation of osmotically active compounds such as hydrophilic proteins from COR/LEA superfamily [8]; monosaccharides and oligosaccharides including sucrose and raffinose family oligosaccharides (RFOs) [37]; polyols; polyamines; etc. All these compounds bind water and stabilise membranes and other proteins upon CA [38]. In cold acclimation studies performed in controlled conditions such as growth chambers, plants are grown under defined photoperiod and temperature regimes which usually represent a continuous cold treatment resulting in an acquisition of maximum frost tolerance. Controlled conditions of growth chambers thus represent an ideal model for the study of the effects of cold on the expressions of cold-inducible *CBF-COR/LEA* genes (pathways). It is assumed that the primary cold signal arises at plasma membrane since low temperatures affect the physical properties and the final fluidity of phospholipid bilayer. Cold perception is probably performed by integral membrane proteins. Two-component histidine kinases with cytoplasmic kinase domain enabling signal transduction into cytoplasm and signal amplification have been proposed to confer cold sensing in cyanobacteria [39,40]. COLD1, identified in rice cv. Nipponbare, encodes a regulator of G protein signalling which is involved in chilling (0–15 °C) sensing and signal transduction via interaction with RGA1, the alpha subunit of a heterotrimeric G protein, and via calcium signalling [41]. An SNP2 in the fourth exon containing allele in COLD1 enhances its activation of G protein signalling, underlying enhanced chilling tolerance in japonica rice grown in temperate regions of Japan [41]. In the nucleus, cold-inducible signalling pathways can generally be divided into two major groups: ABA-dependent and ABA-independent. Under cold, *COR/LEA* gene expression is regulated by both ABA-dependent and ABA-independent pathways such as CBFs, as indicated by the presence of multiple copies of ABRE and CRT/DRE motifs in their promoters [42,43]. Moreover, both ABA-dependent and ABA-independent pathways reveal cross-talk in their regulation since ABA-induced kinase SnRK2 phosphorylates ICE1, which in turn activates *CBF* pathway [44,45]. Continuous cold treatments lead to significantly enhanced expression of *CBF/COR* pathway, indicated by *COR/LEA* transcript peaks at 3–4 days of cold treatment, while a slower rise of corresponding proteins results in a maximum at 2–3 weeks of cold treatment [46,47]. Moreover, organ-related differences in dehydrin protein relative accumulation were found by a comparison of leaf and crown tissues with crowns representing the crucial organ for whole plant survival [47]. Thus, determination of COR/LEA proteins in crown tissues in winter cereals seems to represent a useful approach for assessment of plant winter survival. Due to the high level of cold-induced dehydrin protein accumulation, their direct relationship to plant acquired frost tolerance was repeatedly found [24]. A correlation between dehydrin protein relative accumulation and plant acquired frost tolerance determined as LT_50_ was found not only for winter-type cereals such as winter wheats [25] and barleys [26] but also for other winter-type crops such as winter oilseed rape [48] when grown in controlled environment.

In addition to temperature, the CA process including the expression of some *CBF-COR/LEA* pathway genes and the resulting FT is affected by light including photoperiod, light intensity and light quality, i.e., the red to far-red (R:FR) light ratio [22,49]. Short-day (SD) photoperiods reveal inducing effects on CBF regulon, resulting in an enhanced accumulation of COR/LEA proteins and other cold-protective metabolites such as RFOs [37]. Moreover, relatively higher light irradiances favour CA in comparison to low irradiances since light is necessary for carbon assimilation and enhanced accumulation of protective compounds [50]. Ahres et al., [50] studied barley response to low temperatures (5 and 15 °C) and different light conditions including three light intensities and FR light supplementation on acquired FT and *CBF14*, *COR14b* and *DHN5* transcript levels and found a significant response to both cold and light intensity and FR supplementation for *CBF14* and chloroplast-located *COR14b* genes, while *DHN5* responded only to cold. Studies aimed at a comparison of wheat *Wcs120* or barley *Dhn5* (Kn-type dehydrin) and *Cor14b* (LEA-III) gene expression revealed that *Wcs120/Dhn5* expression pattern correlates well with cold-inducible *CBFs* expression and acquired FT, while the *Cor14b* expression pattern was more different from *CBFs* since it was also regulated by light [49,50].

There are differences between cold-tolerant (winter) and cold-susceptible (spring) genotypes in cereals (wheat and barley) related to *CBF-COR/LEA* pathway induction:(1)Genetic differences: Cold-tolerant winter genotypes encode higher gene copy number of cold-inducible *CBF* genes at *Fr2* locus when compared to cold-susceptible spring ones.The differential induction level of *CBF/COR* pathways between frost-tolerant and frost-susceptible genotypes can be determined genetically; for example, a comparative study by Tondelli et al., [51] revealed that frost-tolerant winter barley Nure has a higher *CBF* gene copy number in cold-inducible *Fr2* locus than frost-susceptible spring barley Tremois. The differences in gene copy number of *CBFs* and other cold-inducible genes can thus underlie the differences in COR/LEA/dehydrin protein accumulation and the resulting frost tolerance between contrasting genotypes (e.g., spring vs. winter ones). Similarly, enhanced levels of *WCBF2* TF and downstream *Cor/Lea* transcripts *Wdhn13, Wcor14* and *Wcor15* were found in frost-tolerant winter wheat Mironovskaya 808 in comparison to frost-susceptible spring wheat Chinese Spring; moreover, the transcript levels of *WCBF2* as well as *Wdhn13, Wcor14* and *Wcor15* peaked later (at 42 days of cold treatment) in Mironovskaya 808 than in Chinese Spring (at 21 days of cold treatment) [52].(2)Threshold induction temperatures: Fowler [29] showed that highly frost-tolerant winter cereals such as rye start inducing enhanced acquired frost tolerance determined as LT_50_ (lethal temperature for 50% of the samples) at higher growth temperatures in comparison with the less tolerant ones. Analogous patterns to LT_50_ were also found for cold-inducible CBFs and COR/LEA proteins, i.e., cold-tolerant winter cultivars start inducing cold-inducible genes such as CBFs and downstream COR/LEA proteins at higher temperatures in comparison with cold-susceptible ones. Vágújfalvi et al., [53] detected *Cor14b* transcripts in frost-tolerant winter line G3116 of einkorn wheat (*T. monococcum*) at higher temperature (up to 20 °C) than in frost-susceptible spring line DV92. Campoli et al., [54] detected different threshold induction temperatures for different *CBF* structural groups based on their phylogenetic analysis in winter barley, winter wheat, two winter rye and one spring rye cultivar. Similarly, Badawi et al., [55] distinguished ten CBF phylogenetic groups in two Triticum species, of which five Pooideae-specific groups revealed higher constitutive and low temperature inducible expression in winter wheat Norstar. Our previous studies [27,28] demonstrated that cold-inducible proteins such as wheat WCS120 or barley DHN5 can be detected in the highly frost-tolerant cultivars such as Mironovskaya 808 or Odesskij 31 at higher temperatures (17–20 °C) than in the less tolerant winter wheats or barleys (around 10 °C).(3)Differential phytohormonal regulation of CA process: CA leads to repression of plant growth and development. In *A. thaliana*, Achard et al., [56] observed a positive effect of *CBF1* on accumulation of DELLA proteins known as growth repressors due to stimulation of GA-2 oxidase resulting in reduction of active gibberellins (GA). Our comparative studies on winter wheat Samanta and spring wheat Sandra, winter wheat Cheyenne-Chinese Spring 5A substitution lines as well as einkorn wheats G3116 (facultative) and DV92 (spring) revealed similar patterns of FT (LT_50_) and dehydrins (COR14b and WCS120 family) induction during the first days of CA treatment; however, at later stages (21–42 days CA), spring genotypes revealed significantly lower FT and WCS120 proteins levels. Phytohormone analyses also revealed a two-phase CA response with an alarm and early acclimation phase (1–3 days CA) with increased ABA in all growth habits inducing stress acclimation response and later CA phases with differential responses (7–42 days) when winter types maintained high levels of stress acclimation-related phytohormones (ABA, JA and SA) while spring types revealed induction of phytohormones involved in vegetative-to-reproductive phase transition such as auxin, bioactive CKs and GAs probably due to *VRN1* gene expression and floral meristem development [57,58,59]. Differential phytohormone dynamics may thus be the reason for lower FT and COR/LEA transcript/protein levels found in spring genotypes compared to winter ones at full CA (2–3 weeks CA treatment).

In the CA process, COR/LEA proteins play important roles of maintaining cell hydration, chaperone and other protective roles in interaction with other proteins and membranes; cryoprotective and antifreeze roles resulting in maintaining enzymatic activities under cold and modification of ice crystal growth under frost, respectively; and radical scavenging and ion binding functions. A scheme summarising the factors determining *COR/LEA* gene expression and their biological functions upon CA treatments is given in Figure 1.

## 3. Effect of Vernalisation on Cold-Inducible Pathways Including Dehydrins

Vernalisation, i.e., the requirement of a sufficiently long period of low temperatures in order to gain potential to flowering, represents an evolutionary adaptation to temperate climates with harsh winter seasons in winter annuals. At molecular level, fulfilment of vernalisation requirement puts an imprint to plant stress memory due to epigenetic modifications of specific vernalisation genes resulting in irreversible downregulation of flowering repressors such as FLC in *A. thaliana* or an activation of flowering inducers such as VRN1 in Triticeae [60]. In Triticeae, fulfilment of vernalisation requirement in cereals winter growth habits leads to a downregulation of the major flowering repressor *VRN2* leading to an induction of *VRN1* which is a MADS-box gene revealing both morphogenetic and transcription regulation functions and acting as a flowering integrator, indicating plant preparedness to flowering [61,62,63]. In Triticeae, upregulation of *VRN1* gene expression is also accompanied by changes in histone methylation status: the level of H3K27me3 (histone 3 lysine 27 trimethylation) decreases and the level of H3K4me3 increases, resulting in an active chromatin state for transcription [64]. Vernalisation studies in winter-type cereals repeatedly confirmed a dominant negative effect of *VRN-1* gene expression on cold-inducible *CBF/COR* pathway expression [4,65], while sthe tudy of vernalisation in model plant *Arabidopsis thaliana* revealed a crucial role of vernalisation-induced *SOC1* gene as a direct inhibitor of *CBF/COR* pathway [66].

Studies on *VRN1/Fr1* substitution lines between winter wheat Norstar and spring wheat Manitou revealed that an induction of *VRN1* gene expression following vernalisation fulfilment correlates with a downregulation of *CBFs* and *COR/LEA* gene expression [4]; however, the studies on Norstar–Manitou substitution lines indicate that the type of *Vrn-A1* locus does not have a significant effect on the initial rates of CBF-COR/LEA pathway gene expression, which is rather governed by the genetic background [67].

There is a negative correlation between the expression of *VRN1* gene expression and *CBF-COR/LEA* cold-inducible pathway. However, the molecular mechanisms underlying VRN1 and CBF-COR/LEA interactions remain to be elucidated since no direct link between *VRN1* and *CBFs* has been identified [68]. Some VRN1 downstream regulated genes were identified in barley by Deng et al., [69] including a flowering promoter *VRN3* gene, also known as *FLOWERING LOCUS T-1 (FT1)* gene, whose expression is activated by VRN1; flowering repressors *VRN2* and *ODDSOC2* genes, whose expressions are downregulated by *VRN1*; and some genes involved in abscisic acid, jasmonic acid and gibberellin biosynthesis pathways. In addition, *VRN1* also seems to positively regulate some morphogenetic genes involved in floral meristem development such as *AP1* [70].

Fulfilment of vernalisation requirement in winter cereals is indicated by induction of *VRN1* gene expression. In *Arabidopsis thaliana*, the link between vernalisation fulfilment and a downregulation of CBF-COR/LEA pathway has already been elucidated. Vernalisation-induced downregulation of a major flowering repressor FLC results in an upregulation of SOC1, a MADS-box flowering integrator which was reported to inhibit CBF gene expression [66].

The study by Li et al., [70] on *Vrn1/Fr1* substitution lines between winter wheat Norstar and spring wheat Manitou (“winter Manitou” and “spring Norstar”) revealed that *Vrn1/Fr1* locus is a dominant but not a sole locus determining acquired frost tolerance expressed as LT_50_ and COR/LEA gene levels since “winter Manitou” has ca 5 °C higher LT_50_ when compared to “Norstar” (and even “spring Norstar” under short days) indicating a joint effect of both *Vrn1/Fr1* locus and the genetic background on the resulting LT_50_ and COR/LEA gene expression. Under short-day conditions which delay vegetative-to-reproductive transition, spring Norstar revealed nearly the same acquired frost tolerance as Norstar due to the same genetic background including predominantly *Fr2* loci harbouring *CBF* cluster on the long arms of group 5 chromosomes and loci on the long arms of group 6 chromosomes encoding *WCS120* genes from dehydrin group [71]. Li et al., [70] detected 21 cold-responsive differentially expressed transcripts including *CBF3*, *WCOR410b*, *CS120*, 3 *CS66* genes highly expressed in the lines with Norstar background, 14 cold-responsive transcripts including *LOS2* and 2 *CS66* genes highly expressed in lines with Manitou background.

The effect of vernalisation fulfilment on the ability of winter wheat Mironovskaya 808 to reacclimate to cold following a period of warmer temperatures inducing deacclimation was studied by Vítámvás and Prášil [72] in controlled conditions. They found that fulfilment of vernalisation significantly reduced winter wheat ability to accumulate WCS120 proteins after reacclimation.

In summary, cold acclimation leads to cessation of plant growth and development aimed to induce enhanced stress tolerance while vernalisation results in an acquisition of the competence to flowering. A comparison of the effects of cold acclimation and vernalisation on winter cereals is provided in Table 1.

## 4. Field Studies: COR/LEA Proteins and Winter Hardiness

Winter seasons in temperate climates can be characterised by a complex of environmental stresses including not only low temperatures (both chilling and freezing temperatures) but also other stress factors such as water regime related stresses including both winter drought as well as temporal flooding events following snow melting, mechanical stress caused by soil heaving, and others. Studies aimed at the effects of multiple stress treatments on plants indicated that the resulting impacts of multiple stresses do not simply equal to the sum of the individual stress treatments thus highlighting the urgent need to study multiple stress impacts in controlled experiments as well as in the field [73]. All these environmental factors determine the final winter hardiness of winter crops. Winter hardiness thus represents a complex phenotypic trait which is usually determined as percentage of plant winter survival. Since different dehydrin proteins reveal a differential response to various environmental stresses, it is thus evident that a sum of various dehydrin proteins detected in the plant samples reveals a better correlation with plant winter survival than a single dehydrin protein [30].

Basically, two major types of temperate climate can be distinguished: a continental climate, which is characterised by continuous freezing temperatures throughout the whole winter, and a maritime /transitional type of climate, which is characterised by fluctuations between freezing and above-zero temperatures that is associated with freezing–thawing cycles of water as well as mechanical stress (damage) caused by ice crystals, soil heaving, etc. Periods of low temperatures are thus interspersed by periods of relatively warm temperatures inducing deacclimation in overwintering plants. It is known that vernalisation fulfilment adversely affects plant ability to reacclimate to low temperatures and newly induce several cold-responsive genes including CBF-COR pathway. Therefore, the level of relative accumulation of several COR/LEA proteins including dehydrins (LEA-II) but also chloroplast-located COR14b (LEA-III) proteins reveals a correlation with plant winter survival only in plants prior to vernalisation fulfilment, i.e., sampled in early winter (until Christmas) [30]. Samplings carried out in later stages (January and February) revealed no significant correlation between plant winter survival and accumulation of COR/LEA proteins [30,74] due to periods of relatively warm temperatures inducing partial deacclimation.

Continental climates are characterised by continuous freezing throughout the whole winter season; thus, enhanced levels of COR/LEA/dehydrin proteins were found during the whole winter season up to March in the crown tissues of winter wheat, triticale and rye [75,76]. Pomortsev et al., [75] found high quantitative levels but differences in qualitative profiles in dehydrins in winter wheat cv. Irkutskaya at different sampling dates (November, February and March) during winter season in Central Siberia, Irkutsk, Russia. The authors detected the presence of 29 kDa dehydrin band only in autumn samplings (November), while no such band was detected in winter and spring samplings (February and March). Ganeshan et al., [76] presented a three-year study on cold acclimation and COR/LEA gene expression in Norstar–Manitou set of substitution lines. The studied COR/LEA (*WCS120*, *WCOR410* and *WCOR14*) transcript levels decreased throughout winter except for *WCOR14* transcript in winter Manitou; however, significant alterations among the different seasons were observed, which corresponded with fluctuations in soil temperature at crown depth.

In contrast, in maritime and transitional zones such as Italy or Czechia, where freezing periods are interspersed by relatively long periods of above-zero temperatures during winter (freeze–thaw cycles) when partial cold deacclimation occurs, relatively low levels of COR/LEA/dehydrin proteins were found in plant tissues in later phases of winter (January and February), and these levels did not correspond with the plant freezing tolerance [30,74] since plants are already vernalised at this time and cannot efficiently re-acclimate after a period of deacclimation.

In Italian studies, a positive relationship between *COR14b* transcript accumulation and plant winter survival was found in samplings from November and December [77] while no significant relationship was found in samplings from January and February [74]. Rizza et al., [78] found no significant relationship between *COR14b* transcript accumulation and acquired frost tolerance in a set of European barley cultivars; in contrast, they proposed Fv/Fm chlorophyll fluorescence parameter as a reliable indicator of acquired FT in barley.

Vítámvás et al., [30] found a positive relationship between cold-inducible dehydrin (sum of WCS120 proteins in winter wheats; DHN5 protein in barleys) accumulation in crown tissues and winter survival in field grown winter wheat and barley plants at two locations (Prague-Ruzyně and Lužany) and two winter seasons in Czechia in December samplings prior to the fulfilment of vernalisation, while the positive relationship between plant winter survival and dehydrin proteins was lost in later samplings (January and February) when plants were already vernalised and exposed to partial deacclimation during warmer periods throughout the winter season.

## 5. Conclusions

The cold acclimation process leading to enhanced low temperature tolerance is associated with profound alterations in gene expression including activation of CBF-COR/LEA pathway. There are significant differences between frost-tolerant and frost-susceptible genotypes at both genotypic and phenotypic levels with frost-tolerant genotypes encoding higher copy numbers of cold-inducible CBF genes as well as revealing higher threshold induction temperatures for cold-inducible genes. Vernalisation fulfilment, associated with an induction of *VRN1* gene, leads to a downregulation of CBF-COR/LEA pathway, thus decreasing the possibility of cold reacclimation following a period of warmer temperatures. For the study of cold acclimation process in controlled conditions (growth chambers), defined environmental conditions such as temperature, photoperiod, irradiance, substrate (soil), fertilisation and plant protection (no pathogens) are used in most experiments. These conditions lead to exact and repeatable results which are surely valuable for plant science research; however, they are very different from those the plants have to face in the field (Table 2).

Winter seasons in temperate climates are characterised not only by low temperatures but a complex of environmental stresses related to plant water regime, mechanical wounding and pathogens. Plant winter hardiness thus represents a complex phenotypic trait affected by various environmental factors and determined by expression of wide arrays of genes. Therefore, rather the sum of all dehydrin proteins than a single protein identified in the plant materials reveals a correlation with winter survival in winter cereals. In temperate climate zone, winter seasons differ between maritime/transitional climatic regions such as western/central Europe and continental climates regions such as central Canada and Siberia. Continental climates are characterised by continuous freezing temperatures, thus overwintering cereals reveal enhanced levels of COR/LEA proteins throughout the whole winter season. In contrast, in maritime/transitional climates characterised by freeze–thaw cycles, high levels of COR/LEA proteins correlating with plant winter survival can be found only prior to vernalisation fulfilment, whereas no significant correlation between COR/LEA protein accumulation and winter survival was found in later phases of winter (January and February) when vernalised plants were exposed to periods of relatively warm temperatures inducing partial deacclimation.

## Figures and Tables

**Figure 1 plants-10-00789-f001:**
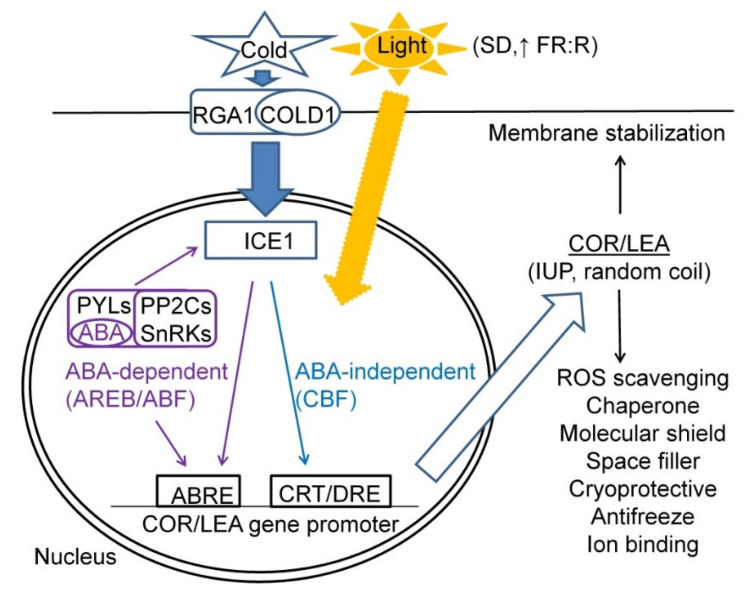
A scheme summarising major factors and signalling pathways regulating *COR/LEA* gene expression and their biological functions in plant cells subjected to CA treatment.

**Table 1 plants-10-00789-t001:** A comparison of cold acclimation and vernalisation processes and their impacts on winter Triticeae.

Characteristics	Cold Acclimation	Vernalisation
Inducing conditions	Short-term cold (days to weeks), Short-day photoperiods	Long-term cold (weeks to months), Long-day photoperiods (VRN3/FT1 pathway)
Plant response	Conservation of vegetative stage; shoot apex: single-ridge (new leaves) High FT induction Phytohormones: ABA, JA, SA (stress tolerance induction), DELLA (growth repressors)	Transition to reproductive stage; shoot apex: double-ridge (floral meristem) Reduced ability to induce FT under LT Phytohormones: auxin, active cytokinins and gibberellins
Gene expression	Upregulation of genes associated with enhanced FT (CBF-COR/LEA) High levels of flowering repressors (VRN2 in winter cereals) are associated with high FT	Downregulation of genes associated with FT acquisition (CBF-COR/LEA) Upregulation of *VRN1* and floral meristem identity genes (*AP1*, *AGL19*, *AGL24*)

**Table 2 plants-10-00789-t002:** A comparison of the impacts of controlled and field conditions on dehydrin relative accumulation and plant tolerance to environment.

Characteristics	Controlled (Growth Chamber)	Field Experiments
Growth conditions	Controlled (growth chamber): a very few variables (usually temperature, or photoperiod)—distinct and contrasting values (e.g., optimum, e.g., +20 °C vs. cold, e.g., +4 °C; long-day 16 h/8 h vs. short-day 8 h/16 h day/night); constant irradiance; defined watering	Very variable, continuously changing conditions with significant fluctuations (temperature) or continuously changing values (photoperiod-day shortening in autumn, day prolongation in spring); several additional stress factors including water-related stress (transient drought or wet-waterlogging and flooding), nutrient-related stress, mechanical wounding, biotic stress (pathogens)
Plant growth stage	Defined: usually early vegetative stage (e.g., 3-leaf stage) or (less often) after vernalisation fulfilment	Continuous development from vegetative to reproductive transition (vernalisation fulfilment)
Plant stress tolerance	Frost tolerance expressed as lethal temperature for 50% samples (LT_50_) determined by laboratory methods (direct frost test) under defined freezing, thawing and recovery conditions	Winter hardiness expressed as percentage of survived plants (winter survival) as a result of joint effects of all stress factors during winter
COR/LEA proteins	A correlation between relative abundance of a single cold-induced COR/LEA protein (a single band on the immunoblots) and LT_50_ both before and after vernalisation fulfilment under continuous cold	A correlation between relative abundance of the sum of cold-inducible COR/LEA proteins (all COR/LEA bands on the immunoblots) and plant winter survival only at early sampling dates prior to vernalisation fulfilment

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
