# Peer review of "COR/LEA Proteins as Indicators of Frost Tolerance in Triticeae: A Comparison of Controlled versus Field Conditions"

_plants, 2021, doi:10.3390/plants10040789_

Round 1

Reviewer 1 Report

The manuscript by Kosová et al., provides a summary of studies dealing with the relationships between plant acquired frost tolerance and COR/LEA transcripts/proteins relative accumulation in cereals grown in controlled and in field conditions. The topic treated by the authors is really interesting and current, especially considering the climatic changes to which our planet is subjected in the recent decades. The authors have well supported the review using recent bibliographic references. However, the review contains very little information to justify its publication as review, so I advise the authors to change the type of publication into a 'mini-review', if possible. On the other hand, the main weakpoint of this review is the lack of figures and diagrams summarizing what it has been reported by the authors. I suggest to introduce summary figures in some sections.

Concerning the structure, the manuscript follows a very logical line. First of all, the state of the art is extensively discussed in the introduction, also describing the potential mechanisms involved. For example, in this section, a table or scheme reporting the classification of the different dehydrins could be added. In the second section, the authors treated the topic evaluating exclusively experimental studies under controlled conditions. Regarding this section, I believe that could be easier for the reader to have a diagram, table, or scheme in which the information described in line 144-198 are reported. Finally the authors treated data obtained from field trials, focusing on the differences between controlled and field conditions in the conclusion section.

Overall, the review is quite interesting and well written.

Author Response

Dear reviewer, 

Thank you very much for your helpful comments. 

We have added a scheme as Figure 1 on signalling and signal transduction leading to Cor/Lea gene expression during the cold acclimation process and a corresponding text to part 2.

For further details, see the attached file. 

Kind regards Klara Kosova

Reviewer 2 Report

The topic of the manuscript COR/LEA proteins as indicators of frost tolerance in Triticeae: A comparison of controlled versus field conditions is interesting for the scientific community, especially cause climate changes, alternation of cold and warm period during winter time are occurring  affecting crop growth.  However the manuscript has two serious concerns  for which it cannot be published in the present version and the second one also  makes useless the review process:

  1. references seem to me  too old and therefore I suggest the authors to update them.
  2.  The manuscript cannot be reviewed before authors clarify why they submit to Plants a manuscript already published in www.preprints.org:

Preprints (www.preprints.org) | NOT PEER-REVIEWED | Posted: 26 February 2021 doi:10.20944/preprints202102.0609.v1

Author Response

Dear reviewer, 

Thank you very much for your comments. 

We have tried to add some novel references to the text and added a new figure with a corresponding text to part 2 on cold acclimation process. 

Regarding the manuscript submission to Preprints, the only reason why we have submitted to Preprints was that it was given as an one-click option during the regular submission process to Plants. We do not understand why Plants provide this option during their regular submission process. 

Now, the text was modified and a new figure was added so the revised manuscript is different from the text in Preprints. 

For further details, see the attached file. 

Kind regards Klara Kosova

Round 2

Reviewer 2 Report

The manuscript has been revised according my previous comments and author replied to my criticism about potential plagiarism.

Therefore the manuscript can be published on Plants after a minor editing of English language, during the publication process. 

Author Response

Dear Reviewer,

thank you very much for your understanding.

We have prepared an original text with two tables which we have submitted only to Plants. However, during the submission process in Plants, I have clicked on the option to submit the manuscript to Preprints which was included as a part of a regular submission process to Plants. We have not performed any parallel submission of the same manuscript to any other journal.

Moreover, now the text was modified, and several references and a new figure were added thus the revised manuscript is different from the version in Preprints.

Kind regards,

Klára Kosová
